# TRAINING-FREE GUIDANCE OF DIFFUSION MODELS FOR GENERALISED INPAINTING

## ABSTRACT

Diffusion models facilitate powerful control over the generative process. Here we introduce training-free guidance, a method for sampling from a broad class of conditional distributions that can be considered generalisations of inpainting. The method is grounded in annealed Langevin dynamics which ensures convergence to the exact conditional distribution, unlike existing methods for inpainting which rely on heuristics. We demonstrate training-free guidance using pretrained unconditional models for image, protein structure, and protein sequence generation and improve upon state-of-the-art approaches. We show the versatility of training-free guidance by addressing a wide range of tasks, including multi-motif scaffolding and amino acid mutagenesis of T cell receptors.

## 1 INTRODUCTION

Denoising diffusion probabilistic models (DDPMs) are a powerful class of generative models (Sohl-Dickstein et al., 2015; Ho et al., 2020) that have gained popularity across many domains (Hoogeboom et al., 2022; Kong et al., 2020). Diffusion models are often trained to generate samples from a data distribution unconditionally, but practical applications generally require sampling from a conditional data distribution (Rombach et al., 2022; Karras et al., 2022).

One such conditional sampling task is inpainting, a well-studied problem where the conditioning specifies the exact values of a subset of the sample. This task is important in several settings, including image editing (Pathak et al., 2016; Liu et al., 2018) and protein engineering, where it is more commonly referred to as motif scaffolding (Didi et al., 2023; Lin et al., 2024). In this work we consider inpainting alongside a suite of other tasks such as editing sequences and optimising an element-wise sample score. These conditions are not typically considered as extensions of inpainting, but we show that they are in fact expressible as a set of inpainting conditions linked by logical connectives (i.e. AND/OR), possibly with varying weights. We refer to these tasks as *generalised inpainting*.

Most methods for conditional sampling require new models (Dhariwal & Nichol, 2021), or modified training procedures (Ho & Salimans, 2022), which can be costly. Several plug-and-play methods have been proposed for both inpainting (Song & Ermon, 2019) and generalised inpainting tasks like sequence editing (Meng et al., 2021), but these methods tend to be heuristically motivated and do not sample from the exact conditional distribution.

In this work we present *training-free guidance* (TFG) of diffusion models, a method of sampling from the exact conditional distribution for generalised inpainting tasks. We show that TFG improves upon the current state-of-the-art for standard inpainting when applied to image and protein structure generation, and give examples of further generalisations of the inpainting problem.

## 2 PRELIMINARIES

### 2.1 DENOISING DIFFUSION MODELS

Given some data distribution $p_0(x)$, where $x \in \mathbb{R}^n$, we can construct a time-dependent family of distributions through the diffusion process

$$dx = -\frac{1}{2}\beta(t)xdt + \sqrt{\beta(t)}dw_t, \tag{1}$$

where $w_t$ is the Wiener process. This yields the family of distributions

$$p_t(x|x_0) = \mathcal{N}\left(x; \sqrt{\bar{\alpha}(t)}x_0, 1 - \bar{\alpha}(t)\right), \tag{2}$$

where $\bar{\alpha}(t) := \exp\int_0^t -\beta(s)ds$. We can then sample from $p_0(x)$ by sampling from $x \sim p_T(x) \approx \mathcal{N}(x; 0, I)$, where $T$ is large, and simulating the reverse-time process given by (Anderson, 1982)

$$dx = \beta(t)\left[-\frac{1}{2}x - \nabla\log p_t(x)\right]dt + \sqrt{\beta(t)}d\bar{w}_t, \tag{3}$$

where $\bar{w}_t$ is the Wiener process with time reversed. In general, the exact score function $\nabla\log p_t(x)$ is unavailable, so an approximation to the score $s_\theta(x_t, t)$ is learnt by minimising the denoising score matching objective (Vincent, 2011)

$$\int_0^\infty \mathbb{E}_{x_0 \sim p_0, x \sim p_t(\cdot|x_0)}\left[\|s_\theta(x, t) - \nabla\log p_t(x|x_0)\|^2\right]dt. \tag{4}$$

In order to sample from the data distribution, DDPMs (Sohl-Dickstein et al., 2015; Ho et al., 2020) simulate Equation 3 using the discretisation

$$x_{t-\Delta t} = \frac{x_t + \beta_t s_\theta(x_t, t)}{\sqrt{1-\beta_t}} + \sqrt{\beta_t}z_t, \tag{5}$$

where $\beta_t := \beta(t)\Delta t$ and $z_t \sim \mathcal{N}(0, I)$.

## 2.2 Annealed Langevin Dynamics

The diffusion process yields a family of distributions $p_t(x)$ that anneal to the data distribution $p_0(x)$. Sampling from an annealed family in order to eventually sample from $p_0(x)$ has a history that predates generative diffusion models (Kirkpatrick et al., 1983; Neal, 2001). A popular method involves Langevin dynamics (Parisi, 1981), which relies on the fact that the stochastic process

$$dx = \nabla\log p_t(x)d\tau + \sqrt{2}dw_\tau \tag{6}$$

has the stationary distribution $p_t(x)$. Therefore we can simulate a discretisation of this process to sample from $p_t(x)$ for each timestep $t$, and use this value as the initialisation for dynamics at the next timestep $t - \Delta t$.

## 3 Theory

### 3.1 Inpainting and the Problem with Replacement Sampling

Inpainting is the problem of sampling from some data distribution $p_0(x)$, where $x \in \mathbb{R}^n$, conditioned on a subset of dimensions $M$ being fixed to a target value, $p_0(x|x^{\in M} = \tilde{x})$. An intuitive approach to inpainting, which we refer to as replacement sampling, involves evaluating the score at each timepoint with the value of the dimension replaced with the appropriately rescaled target value (Song et al., 2021). That is, the reverse-time process is simulated via

$$x_{t-\Delta t}^{\notin M} = \frac{x_t^{\notin M} + \beta_t s_\theta\left(x_t^{\notin M} \oplus \sqrt{\bar{\alpha}(t)}\tilde{x}, t\right)^{\notin M}}{\sqrt{1-\beta_t}} + \sqrt{\beta_t}z_t^{\notin M}. \tag{7}$$

The original formulation of replacement sampling also involves the addition of noise, the details of which do not affect the results of this section but can be found in Appendix A.

By comparison with Equation 5, we see that Equation 7 would be the discretisation of the reverse-time process with the family of distributions

$$p_t^{\text{replace}}(x^{\notin M}) := p_t\left(x^{\notin M} \oplus \sqrt{\bar{\alpha}(t)}\tilde{x}\right)^{\notin M}. \tag{8}$$

However, this family does not correspond to a forward diffusion process, and therefore the argument of Anderson (1982) that leads to Equation 3 cannot be applied. To see this, recall that $p_t(x)$ corresponds, by definition, to the forward diffusion process, and as such satisfies the relevant Fokker-Planck equation (Fokker, 1914; Planck, 1917),

$$\frac{\partial}{\partial t}p_t - \frac{1}{2}\nabla \cdot (xp_t) - \frac{1}{2}\nabla^2 p_t = 0. \tag{9}$$

By contrast, the left-hand side of the relevant Fokker-Planck equation for $p_t^{\text{replace}}(x^{\notin M})$ reads

$$\frac{\partial}{\partial t}p_t^{\text{replace}} - \frac{1}{2}\nabla_{\notin M} \cdot (x^{\notin M} p_t^{\text{replace}}) - \frac{1}{2}\nabla_{\notin M}^2 p_t^{\text{replace}}$$

$$= \left(\frac{\partial}{\partial t}p_t - \frac{1}{2}\nabla \cdot (xp_t) - \nabla^2 p_t\right) + \frac{1}{2}\nabla_{\in M} \cdot (x^{\in M} p_t) + \frac{1}{2}\nabla_{\in M}^2 p_t$$

$$= \frac{1}{2}\nabla_{\in M} \cdot (x^{\in M} p_t) + \frac{1}{2}\nabla_{\in M}^2 p_t, \tag{10}$$

which does not vanish in general.

## 3.2 A Toy Problem

Even in very simple settings, replacement sampling can fail drastically. Consider the joint distribution, plotted in Figure 1, given by

$$p_0(x, y) = \begin{cases} 0.9 & \text{if } x = +1 \text{ and } y = +1, \\ 0.09 & \text{if } x = -1 \text{ and } y = -1, \\ 0.01 & \text{if } x = +1 \text{ and } y = -1, \end{cases} \tag{11}$$

and suppose we want to sample from the conditional distribution

$$p_0(x|y = -1) = \begin{cases} 0.9 & \text{if } x = -1, \\ 0.1 & \text{if } x = +1. \end{cases} \tag{12}$$

Empirically, we see in Figure 1 that replacement sampling of this conditional distribution is plagued by the influence of the marginal distribution

$$p_0(x) = \begin{cases} 0.09 & \text{if } x = -1, \\ 0.91 & \text{if } x = +1. \end{cases} \tag{13}$$

Moreover, it is also clear from this experiment that this issue cannot be rectified by simply increasing the number of sampling steps; the replacement sampling method inevitably converges to the wrong distribution. Indeed, we show explicitly that the Fokker-Planck equation is not satisfied for this distribution in Appendix B, so replacement sampling does not sample from the conditional distribution.

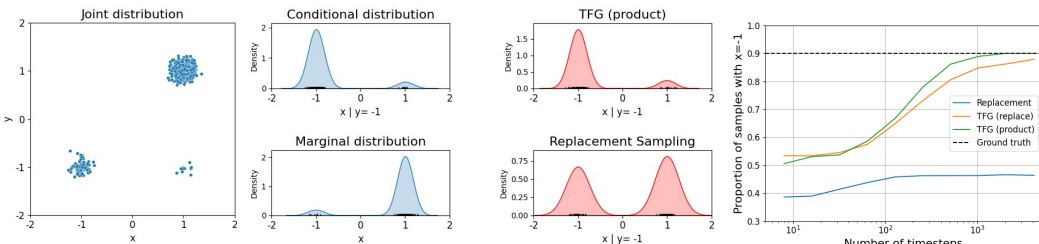

Figure 1: Replacement sampling fails in a toy example in which the conditional distribution is significantly different to the marginal distribution. By contrast, the samples generated from TFG with both $p_t^{\text{replace}}(x)$ and $p_t^{\text{product}}(x)$ match the conditional distribution, given sufficient number of timesteps.

## 3.3 A Solution to the Aforementioned Problem

For simplicity of notation, we consider the case where only the final of the $n$ dimensions is fixed to a target $\tilde{x}$. That is, consider sampling from $p_0(x|x^{(n)} = \tilde{x})$. Notice that although the family of distributions $p_t^{\text{replace}}(x^{(1:n-1)})$ does not correspond to a forward diffusion process, it *does anneal* to the desired distribution $p_0(x|x^{(n)} = \tilde{x})$, so we are at liberty to use annealed MCMC approaches. This leads us to the simplest form of TFG, where we use annealed Langevin dynamics (Parisi, 1981) to sample from $p_t^{\text{replace}}(x)$, as presented in Algorithm 1.

Annealed Langevin dynamics is a familiar technique for the sampling of unconditional DDPMs, and is inherent to the predictor-corrector framework (Song et al., 2021). The key insight that enables us to generalise TFG to a much wider range of tasks is that we do not have to use $p_t^{\text{replace}}(x^{(1:n-1)})$ as our annealed family. Instead, we can define any family of distributions – including a family that can be adapted for generalised inpainting problems. To this end, we define a new family of distributions as the normalised product of the unconditional diffusion process and a condition-forcing term:

$$p_t^{\text{product}}(x) := \frac{1}{Z_t} p_t(x) \times \mathcal{N}\left(x^{(n)}; \sqrt{\bar{\alpha}(t)}\tilde{x}, 1 - \bar{\alpha}(t)\right). \tag{14}$$

In Algorithm 2 we show the procedure for TFG with $p_t^{\text{product}}(x)$. Revisiting our toy problem, Figure 1 shows that applying either Algorithm 1 or 2 samples from the conditional distribution correctly provided one takes sufficiently many timesteps.

---

**Algorithm 1** Training-free guidance for inpainting using $p_t^{\text{replace}}(x)$

---

**Require:** target value $\tilde{x}$, step size $\eta$, number of inner loop iterations $N_{\text{inner}}$
1: $x_T \sim \mathcal{N}(0, I)$
2: **for** $t = T, \ldots, \Delta t$ **do**
3:     $\eta(t) \leftarrow \eta\sqrt{1 - \bar{\alpha}(t)}$
4:     **for** $u = 1, \ldots, N_{\text{inner}}$ **do**
5:       $z \sim \mathcal{N}(0, I)$
6:       $x_t \leftarrow x_t + \eta(t)s_\theta\left(x_t^{(1:n-1)} \oplus \sqrt{\bar{\alpha}(t)}\tilde{x}, t\right) + z\sqrt{2\eta(t)}$
7:     **end for**
8:     $x_{t-\Delta t} \leftarrow x_t$
9: **end for**
10: **return** $x_0$

---

---

**Algorithm 2** Training-free guidance for inpainting using $p_t^{\text{product}}(x)$

---

**Require:** target value $\tilde{x}$, step size $\eta$, number of inner loop iterations $N_{\text{inner}}$
1: $x_T \sim \mathcal{N}(0, I)$
2: **for** $t = T, \ldots, \Delta t$ **do**
3:     $\eta(t) \leftarrow \eta\sqrt{1 - \bar{\alpha}(t)}$
4:     **for** $u = 1, \ldots, N_{\text{inner}}$ **do**
5:       $z \sim \mathcal{N}(0, I)$
6:       $x_t \leftarrow x_t + \eta(t)\left[s_\theta(x_t, t) - \left(0 \oplus \cdots \oplus 0 \oplus \frac{x_t^{(n)} - \sqrt{\bar{\alpha}(t)}\tilde{x}}{1 - \bar{\alpha}(t)}\right)\right] + z\sqrt{2\eta(t)}$
7:     **end for**
8:     $x_{t-\Delta t} \leftarrow x_t$
9: **end for**
10: **return** $x_0$

---

## 3.4 Extension to Generalised Inpainting

Let us now turn our attention towards generalised inpainting conditions, which can be expressed as logical combinations of inpainting conditions. For each such task there is a corresponding condition-forcing term, which can be obtained by replacing AND connectives ($\wedge$) with a relevant product of

distributions and `OR` connectives ($\vee$) with a relevant sum of distributions. We state some examples of this procedure in this section and list the corresponding score functions in Appendix D.

**Floating inpainting** Suppose we want to fix $N_{\text{fix}} = 1$ dimensions to some target value $\tilde{x}$, but that we do not know which dimensions to fix. This can be posed as a composition of inpainting tasks via $(x^{(1)} = \tilde{x}) \vee (x^{(2)} = \tilde{x}) \vee \ldots \vee (x^{(n)} = \tilde{x})$. This corresponds to the distribution

$$p_t^{\text{float}}(x) := \frac{1}{Z_t} p_t(x) \times \sum_{i=1}^{n} \mathcal{N}\left(x^{(i)}; \sqrt{\bar{\alpha}(t)}\tilde{x}, 1 - \bar{\alpha}(t)\right). \tag{15}$$

**Sequence editing** Suppose we start with a sample $\tilde{x}$ and want to generate a new sample with at most $N_{\text{edit}} = 1$ dimensions changed, but we do not know which dimensions to edit. This can be posed as a composition of inpainting tasks via $((x^{(2)} = \tilde{x}^{(2)}) \wedge (x^{(3)} = \tilde{x}^{(3)}) \wedge \ldots \wedge (x^{(n)} = \tilde{x}^{(n)})) \vee ((x^{(1)} = \tilde{x}^{(1)}) \wedge (x^{(3)} = \tilde{x}^{(3)}) \wedge \ldots \wedge (x^{(n)} = \tilde{x}^{(n)})) \vee \ldots \vee ((x^{(1)} = \tilde{x}^{(1)}) \wedge (x^{(2)} = \tilde{x}^{(2)}) \wedge \ldots \wedge (x^{(n-1)} = \tilde{x}^{(n-1)}))$. This corresponds to the distribution

$$p_t^{\text{perturb}}(x) := \frac{1}{Z_t} p_t(x) \times \sum_{i=1}^{n} \prod_{j \neq i} \mathcal{N}\left(x^{(j)}; \sqrt{\bar{\alpha}(t)}\tilde{x}^{(j)}, 1 - \bar{\alpha}(t)\right). \tag{16}$$

**Element-wise score optimisation** Suppose that each of the possible values of a dimension $\{\tilde{x}_j\}$ can be assigned a corresponding score $\{w_j\}$, and we want to generate samples such that the total sum of the scores in each dimension is controllable. This can be posed as a composition of inpainting tasks via $((w_1(x^{(1)} = \tilde{x}_1) \vee w_2(x^{(1)} = \tilde{x}_2) \vee \ldots \vee w_j(x^{(1)} = \tilde{x}_j)) \wedge \ldots \wedge (w_1(x^{(n)} = \tilde{x}_1) \vee w_2(x^{(n)} = \tilde{x}_2) \vee \ldots \vee w_j(x^{(n)} = \tilde{x}_j))$. This corresponds to the distribution

$$p_t^{\text{linear}}(x) := \frac{1}{Z_t} p_t(x) \times \mu \prod_{i=1}^{n} \sum_{j} w_j \mathcal{N}\left(x^{(i)}; \sqrt{\bar{\alpha}(t)}\tilde{x}_j, 1 - \bar{\alpha}(t)\right), \tag{17}$$

where varying $\mu$ enables control of the score.

**Element-wise mean score optimisation** Suppose now that the sample is of variable length, and that we want to control the mean score rather than the total score. In this case, we should divide the weighting assigned to a sample from the distribution through by the length of that sample. This corresponds to the distribution

$$p_t^{\text{mean}}(x) := \frac{1}{Z_t} p_t(x) \times \frac{\mu \prod_{i=1}^{n} \sum_{j} w_j \mathcal{N}\left(x^{(i)}; \sqrt{\bar{\alpha}(t)}\tilde{x}_j, 1 - \bar{\alpha}(t)\right)}{\prod_{i=1}^{n} \sum_{j \neq \text{PAD}} \mathcal{N}\left(x^{(i)}; \sqrt{\bar{\alpha}(t)}\tilde{x}_j, 1 - \bar{\alpha}(t)\right)}. \tag{18}$$

### 3.5 INTERLUDE: THE SURPRISING BUT ENTIRELY REASONABLE EFFECTIVENESS OF REPAINT

RePaint is an extension of replacement sampling that leads to a marked improvement in sample quality (Lugmayr et al., 2022). Instead of performing one forward pass at each timepoint, RePaint repeatedly denoises the sample using Equation 7 and then renoises it via

$$x_t = \sqrt{1 - \beta_{t-\Delta t}} x_{t-\Delta t} + \beta_{t-\Delta t} z, \tag{19}$$

where $z \sim \mathcal{N}(0, I)$. Having demonstrated the problem with replacement sampling in Section 3.1, it may seem surprising that RePaint is able to provide significant improvements over replacement sampling, as it also appears to rely on the reverse-time diffusion process. However, in Appendix C we show that in the limit of small $\Delta t$, RePaint performs the update

$$x_t \leftarrow x_t + \beta(t)\Delta t \nabla \log p_t^{\text{repaint}}(x_t) + \sqrt{2\beta(t)\Delta t} z, \tag{20}$$

where $p_t^{\text{repaint}}(x_t)$ is a family of distributions that anneals to $p_0(x|x^{(n)} = \tilde{x})$. Notice that despite the heuristic motivation for RePaint, in this limit we recapitulate a similar form to TFG framework for $p_t^{\text{repaint}}(x)$, but with step size $\beta(t)\Delta t$.

## 4 EXPERIMENTS

### 4.1 INPAINTING

**Images** We first demonstrate TFG on the standard inpainting task using a pretrained unconditional DDPM on the CIFAR10 dataset (Krizhevsky et al., 2009). We consider four standard inpainting tasks, which we denote Left, Top, Inner, and Outer, corresponding to the portion of the image provided to the model. We compare TFG to unconditional generation, replacement sampling (Song et al., 2021), manifold constrained gradients (MCG) (Chung et al., 2022), and RePaint (Lugmayr et al., 2022). For implementation details, refer to Appendix F. We calculate the mean squared error (MSE) and Learned Perceptual Image Patch Similarity (LPIPS) (Zhang et al., 2018) to assess the similarity between the original and reconstructed images. TFG outperforms baseline methods with $p_t^{\text{replace}}(x)$ and $p_t^{\text{product}}(x)$ performing similarly. We show two examples in Figure 2.

| | Left | | Top | | Inner | | Outer | |
|---|---|---|---|---|---|---|---|---|
| | MSE ($\downarrow$) | LPIPS ($\downarrow$) | MSE ($\downarrow$) | LPIPS ($\downarrow$) | MSE ($\downarrow$) | LPIPS ($\downarrow$) | MSE ($\downarrow$) | LPIPS ($\downarrow$) |
| Unconditional | 0.240±0.002 | 0.107±0.001 | 0.220±0.002 | 0.111±0.002 | 0.377±0.006 | 0.134±0.002 | 0.106±0.001 | 0.104±0.002 |
| Replacement | 0.169±0.003 | 0.072±0.001 | 0.187±0.003 | 0.078±0.001 | 0.323±0.006 | 0.108±0.001 | 0.066±0.001 | 0.059±0.001 |
| MCG | 0.119±0.002 | 0.061±0.001 | 0.157±0.004 | 0.078±0.001 | 0.251±0.005 | 0.097±0.001 | 0.054±0.001 | 0.059±0.001 |
| RePaint | 0.124±0.003 | 0.056±0.001 | 0.167±0.003 | 0.075±0.001 | 0.252±0.004 | 0.081±0.001 | 0.040±0.001 | **0.040±0.001** |
| TFG ($p_t^{\text{replace}}$) | 0.100±0.001 | **0.047±0.001** | 0.136±0.006 | **0.065±0.002** | **0.171±0.004** | **0.063±0.001** | **0.038±0.001** | 0.041±0.001 |
| TFG ($p_t^{\text{product}}$) | **0.099±0.002** | 0.052±0.000 | **0.116±0.002** | 0.067±0.001 | 0.178±0.004 | 0.073±0.001 | **0.038±0.001** | **0.040±0.001** |

Table 1: Quantitative comparison between different sampling methods for inpainting on CIFAR10.

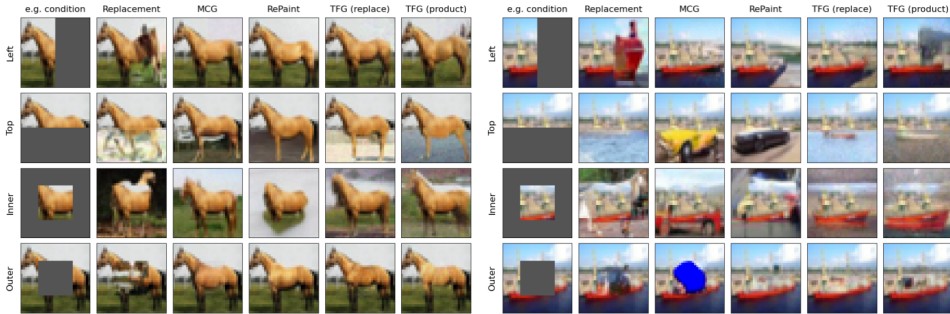

Figure 2: Qualitative comparison between different sampling methods for inpainting on CIFAR10.

Both TFG and RePaint require iterating over $N_{\text{inner}}$ inner loop steps at each timestep. In Table 1 we fix $N_{\text{inner}} = 10$, and demonstrate the effect of varying $N_{\text{inner}}$ in Figure 3 (left). The perceptual quality of RePaint generated images deteriorates at greater values of $N_{\text{inner}}$ whereas TFG continues to improve. In Figure 3 (right) we show that image quality also improves at increased values of temperature, with the effect more signficant for TFG than replacement sampling and RePaint. We hypothesise that is due to the fact that multiplying the score by a constant factor does not sample from the tempered distribution when using the reverse-time diffusion process, as we demonstrate in Appendix E.

**Proteins** In protein engineering, keeping a structural motif fixed and using a generative model to propose a supporting scaffold is a key technique for preservation and optimisation of protein function. To assess TFG in this setting, we use the unconditional version of RFDiffusion (Watson et al., 2023), and evaluate performance on their set of 25 scaffolding tasks. We report the number of *successes* and *unique successes*, defined by Lin et al. (2024) to be the number of sequences that satisfy a set of spatial and model confidence criteria upon refolding sequences derived from generated backbones. Details of this benchmark can be found in Appendix G. Figure 4 shows that TFG once again outperforms baseline methods. We see that $p_t^{\text{replace}}(x)$ achieves more successes, although $p_t^{\text{product}}(x)$ performs comparably when considering the diversity of results.

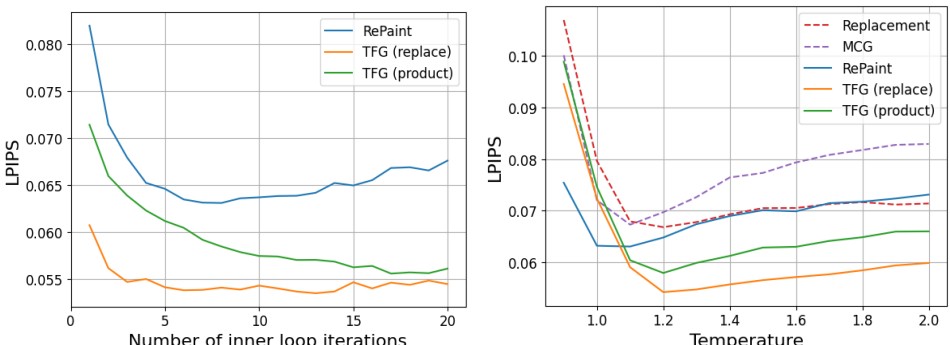

Figure 3: (Left) Increasing the number of inner loops to $N_{\text{inner}} = 20$ improves perceptual quality for TFG, but not for RePaint. (Right) Perceptual image quality improves for temperatures greater than one.

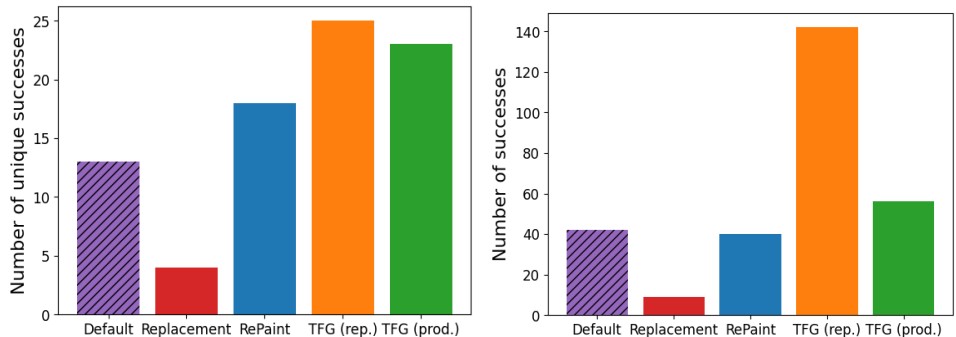

Figure 4: TFG generates more successful samples than other baselines. Although sampling with $p_t^{\text{replace}}(x)$ produces many more total successful scaffolds than $p_t^{\text{product}}(x)$, it generates a comparable number of unique successful scaffolds after clustering similar structures.

## 4.2 FLOATING INPAINTING

A natural extension of the standard inpainting task is *floating inpainting*, whereby a subset of target values are specified but their position is allowed vary within the sample (Liu et al., 2024). Existing methods for sampling necessarily require a position to be fixed prior to sampling. In this section we demonstrate that this relative positional knowledge is already encoded in the diffusion model and can be extracted with TFG.

**Images** We consider the case in which a model is conditioned to include one quadrant of an image of a face – top left (TL), top right (TR), bottom right (BR), or bottom left (BL) – but no information about which of the four quadrants the patch originates from is provided. We use a DDPM pretrained on the CelebA dataset (Liu et al., 2015) and calculate the proportion of samples for which TFG samples the correct quadrant for 1000 images from the test set. As shown in Figure 5, TFG with $p_t^{\text{float}}(x)$ both correctly assigns the correct patch location and preserves the perceptual quality of the inpainted image. Further details can be found in Appendix H.

**Proteins** It is valuable in protein design to be able to generate scaffolds given multiple motifs. To demonstrate TFG in this case, we consider a protein with two motifs, of length six and fourteen amino acids, respectively. We scaffold the multi-motif using the same unconditional model setup as in Section 4.1. We compare the quality of structures generated by TFG with $p_t^{\text{float}}(x)$ with the structures generated by replacement sampling and TFG with $p_t^{\text{float}}(x)$. In the latter two cases, the number of amino acids between the two motifs is chosen randomly prior to sampling, whereas no *a priori* choice is required for TFG with $p_t^{\text{float}}(x)$. The quality of the resulting structures is measured by the predicted Local Distance Difference Test output of the denoising model (pLDDT$_{\text{RF}}$), and

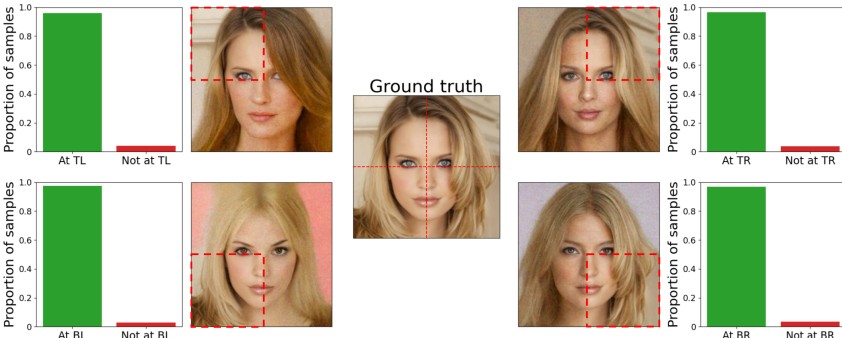

Figure 5: TFG with $p_t^{\text{float}}(x)$ selects the correct quadrant in which to place an image patch in the vast majority of cases. We show a particular example of the results of TFG that arise from conditioning from the same image for each of the four quadrants.

an illustrative example for each method is shown in Figure 6. Our results show that allowing the sampling method to choose the relative position of the fixed motifs, via $p_t^{\text{float}}(x)$, over the course of the sampling process results in an improvement to protein generation. Further details can be found in Appendix K.

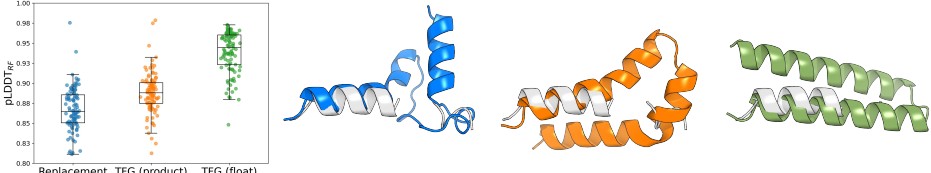

Figure 6: (Left) Using TFG on $p_t^{\text{float}}(x)$ results in better scaffold design than choosing the number of intervening amino acids prior to sampling with replacement sampling or TFG with $p_t^{\text{product}}(x)$. (Right) Example structures conditioned on the same multi-motif (white), sampled using replacement sampling (blue), TFG with $p_t^{\text{product}}(x)$ (orange), and TFG with $p_t^{\text{float}}(x)$ (green).

### 4.3 SEQUENCE EDITING

Sequence editing is a generalised inpainting problem that involves altering an existing sample so that the result is no more than $N_{\text{edit}}$ edits away from the original.

**Proteins** Amino acid mutagenesis is a problem in sequence-level protein design that requires taking a protein and identifying amino acids to mutate. The crucial challenge here is that the positions to mutate are unknown *a priori*. SDEdit (Meng et al., 2021) is a popular method to address this challenge (Vázquez Torres et al., 2024). It involves partially noising the sample, and then using the diffusion model to remove the noise. In Section 3.4 we show that this task can be addressed by TFG with the family $p_t^{\text{perturb}}(x)$ given in Equation 16.

We perform amino acid mutagenesis on the third complementarity determining region on the T cell receptor beta chain (CDR3$\beta$), a highly variable span of protein sequence that mediates cell-mediated immune system activation (Murphy & Weaver, 2016; Want et al., 2023). Modifying CDR3$\beta$ is a key strategy for designing new T cell receptor-based therapies, and operating in sequence space is advantageous since the structure of this region is highly flexible.

We train a DDPM on a large set of CDR3$\beta$ chains (refer to Appendix J for details on the model architecture and training). To validate TFG, we take several batches of 1000 CDR3$\beta$ sequences from the test set and for each sequence generate a new sequence that is at most $N_{\text{edit}} = 1$ edit distance away from the original. A likelihood measure of CDR3$\beta$ sequences can be calculated for the batch before and after mutagenesis using OLGA (Sethna et al., 2019). In addition to our

comparison of TFG to SDEdit, we demonstrate the value of using TFG with $p_t^{\text{perturb}}(x)$ over naively selecting the position to mutate ahead of sampling with TFG on $p_t^{\text{replace}}(x)$. We use two strategies for selecting the mutation position: sampling in proportion to the entropy of the data distribution at that position, and sampling the position uniformly randomly. The results are presented in Figure 7 (left) and show that TFG with $p_t^{\text{perturb}}(x)$ outperforms all other baselines.

### 4.4 ELEMENT-WISE SCORE OPTIMISATION

A further task that can be considered generalised inpainting is the optimisation of a score that can be calculated as the mean of the scores of individual components of a sample.

**Proteins** Hydrophobicity is a property of proteins that plays a crucial role in protein folding (Dobson, 2003). The Kyte-Doolittle (KD) scale (Kyte & Doolittle, 1982) is a measure of hydrophobicity that assigns a numerical value to the twenty different amino acids. To optimise the mean KD score of a protein sequence, we can sample using the distribution $p_t^{\text{mean}}(x)$ with varying values of $\mu$, as defined in Equation 18. To demonstrate control over hydrophobicity, we sample several batches of 1000 CDR3$\beta$s with different values of $\mu$. The results in Figure 7 (right) demonstrate that we are able to control the mean KD score of generated CDR3$\beta$. Further details can be found in Appendix K.

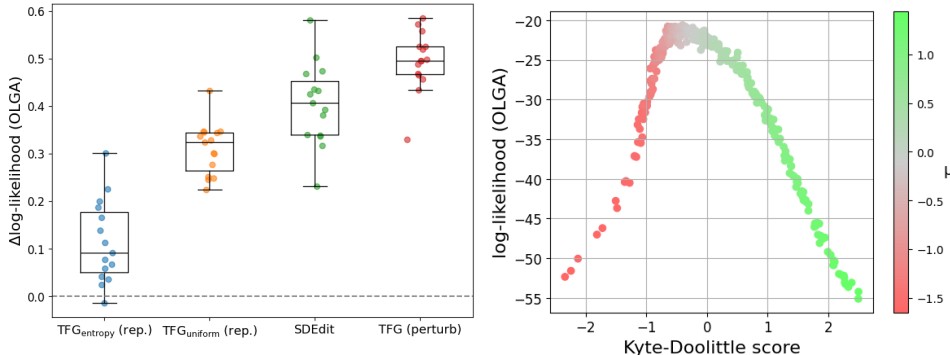

Figure 7: (Left) TFG with $p_t^{\text{perturb}}(x)$ increases the likelihood of a perturbed batch of CDR3$\beta$s more than SDEdit. Moreover, comparison with sampling using $p_t^{\text{replace}}(x)$ demonstrates the importance of allowing the sampler to choose the mutated position, rather than naively choosing it ahead of sample time. (Right) The hydrophobicity of a generated batch of CDR3$\beta$s can be controlled using $p_t^{\text{mean}}(x)$.

## 5 LIMITATIONS AND FURTHER WORK

In this work we have developed training-free guidance, a new method of exact sampling for generalised inpainting conditions. We now highlight some limitations to the approach and suggest possible avenues for further research.

We have restricted ourselves here to Langevin dynamics as an approach for annealed MCMC. The sampling time scales linearly with the number of Langevin steps, which limits the applicability of TFG in scenarios where speed of generation is critical. Improving the efficiency of MCMC methods is a rich field (Andrieu et al., 2010; Neal, 2012) with direct implications for such contexts.

Despite the broad nature of generalised inpainting tasks, they do not comprise all interesting conditional tasks, and we give two possible directions of further generalisation. First, generalised inpainting tasks are straightforward functions in the original data space, not a latent space. Future work could explore the applicability of TFG to conditioning on preferred values in latent spaces, with consequences for latent diffusion models (Rombach et al., 2022). Second, we have not explored condition-forcing terms that cannot be expressed solely in terms of the score, but also require estimates of the raw probabilities $p_t(x)$. These probabilities can be accessed by training energy-based diffusion models (Du et al., 2023), enabling sampling from a wider range of conditions.

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

## A    REPLACEMENT SAMPLING WITH NOISE

In most treatments of replacement sampling, noise is added to the known region after scaling it. That is, rather than updating the sample using (via a learnt approximation)

$$x_{t-\Delta t}^{\notin M} = \frac{x_t^{\notin M} + \beta_t \nabla \log p_t \left( x_t^{\notin M} \oplus \sqrt{\bar{\alpha}(t)}\tilde{x} \right)}{\sqrt{1-\beta_t}} + \sqrt{\beta_t} z_t^{\notin M}, \tag{21}$$

we have

$$x_{t-\Delta t}^{\notin M} = \frac{x_t^{\notin M} + \beta_t \nabla \log p_t \left( x_t^{\notin M} \oplus \sqrt{\bar{\alpha}(t)}\tilde{x} + \sqrt{1-\bar{\alpha}(t)}z_t^{\in M} \right)}{\sqrt{1-\beta_t}} + \sqrt{\beta_t} z_t^{\in M}, \tag{22}$$

where at each timestep we draw $z_t^{\in M} \sim \mathcal{N}(0, I)$ and $z_t^{\notin M} \sim \mathcal{N}(0, I)$ independently. In the limit $\Delta t \to 0$, the gradient step approaches

$$\mathbb{E}_{z \sim \mathcal{N}(0,I)} \left[ \nabla \log p_t \left( x_t^{\notin M} \oplus \sqrt{\bar{\alpha}(t)}\tilde{x} + \sqrt{1-\bar{\alpha}(t)}z \right) \right], \tag{23}$$

so that

$$x_{t-\Delta t}^{\notin M} = \frac{x_t^{\notin M} + \beta_t \nabla \log p_t^{\text{replace}*}(x_t^{\notin M})}{\sqrt{1 - \beta_t}} + \sqrt{\beta_t} z_t^{\notin M}, \tag{24}$$

where

$$p_t^{\text{replace}*}(x^{\notin M}) := \frac{1}{Z_t} \exp \mathbb{E}_{z \sim \mathcal{N}(0,I)} \left[ \log p_t \left( x_t^{\notin M} \oplus \sqrt{\bar{\alpha}(t)} \tilde{x} + \sqrt{1 - \bar{\alpha}(t)} z \right) \right] \tag{25}$$

is the implied family of distributions that the backward process obeys, where $Z_t$ are the appropriate normalisation factors. To see that adding noise into the replacement process does not solve the problem with replacement sampling by satisfing the relevant Fokker-Planck equation, we note that only the dimensions $x \in M$ depend on $z$, so derivatives with respect to $t$ and $x \notin M$ can be moved inside the expectation, and

$$\frac{\partial}{\partial t} p_t^{\text{replace}*} - \frac{1}{2} \nabla_{\notin M} \cdot (x^{\notin M} p_t^{\text{replace}*}) - \frac{1}{2} \nabla_{\notin M}^2 p_t^{\text{replace}*}$$

$$= p_t^{\text{replace}*} \times \mathbb{E} \left[ \frac{\partial}{\partial t} \log p_t - \frac{1}{2} \nabla_{\notin M} \cdot (x^{\notin M} \log p_t) - \frac{1}{2} \nabla_{\notin M}^2 \log p_t \right]$$

$$= p_t^{\text{replace}*} \times \mathbb{E} \left[ \frac{1}{p_t} \left( \frac{1}{2} \nabla_{\in M} \cdot (x^{\in M} p_t) + \frac{1}{2} \nabla_{\in M}^2 p_t \right) \right]. \tag{26}$$

The term inside the innermost bracket is now identical to the non-vanishing term we found when performing the same derivation for replacement sampling without noise, and we recover the same issue with replacement sampling.

## B  MATHEMATICAL DETAILS OF THE TOY EXAMPLE

The forward diffusion process

$$dx = -\frac{1}{2} \beta(t) x dt + \sqrt{\beta(t)} dw_t \tag{27}$$

applied to the initial distribution

$$p_0(x, y) = 0.9\delta(x - 1)\delta(y - 1) + 0.09\delta(x + 1)\delta(y + 1) + 0.01\delta(x - 1)\delta(y + 1) \tag{28}$$

leads to the family of distributions

$$p_t(x, y) = \frac{1}{\sqrt{2\pi(1 - \bar{\alpha}(t))}} \left[ 0.9 \exp\left( -\frac{(x - \sqrt{\bar{\alpha}(t)})^2 + (y - \sqrt{\bar{\alpha}(t)})^2}{2(1 - \bar{\alpha}(t))} \right) \right.$$

$$+ 0.09 \exp\left( -\frac{(x + \sqrt{\bar{\alpha}(t)})^2 + (y + \sqrt{\bar{\alpha}(t)})^2}{2(1 - \bar{\alpha}(t))} \right)$$

$$\left. + 0.01 \exp\left( -\frac{(x - \sqrt{\bar{\alpha}(t)})^2 + (y + \sqrt{\bar{\alpha}(t)})^2}{2(1 - \bar{\alpha}(t))} \right) \right]. \tag{29}$$

In the toy example in Section 3.2, we look to sample the conditional distribution with the $y$-dimension fixed to $-1$ by considering

$$p_t^{\text{replace}}(x) = p_t(x, -\sqrt{\bar{\alpha}}(t))$$

$$= \frac{1}{\sqrt{2\pi(1 - \bar{\alpha}(t))}} \left[ \left( 0.9 e^{-2\bar{\alpha}/(1-\bar{\alpha})} + 0.01 \right) \exp\left( -\frac{(x - \sqrt{\bar{\alpha}(t)})^2}{2(1 - \bar{\alpha}(t))} \right) \right.$$

$$\left. + 0.09 \exp\left( -\frac{(x + \sqrt{\bar{\alpha}(t)})^2}{2(1 - \bar{\alpha}(t))} \right) \right]. \tag{30}$$

Now

$$\frac{\partial}{\partial t}p_t^{\text{replace}} - \frac{1}{2}\nabla_{\notin M} \cdot (x^{\notin M}p_t^{\text{replace}}) - \frac{1}{2}\nabla_{\notin M}^2 p_t^{\text{replace}}$$

$$= \frac{1}{2}\nabla_{\in M} \cdot (x^{\in M}p_t) + \frac{1}{2}\nabla_{\in M}^2 p_t$$

$$= \frac{1}{2}\frac{\partial}{\partial y}(yp_t) + \frac{1}{2}\frac{\partial^2 p_t}{\partial y^2}$$

$$= \frac{1}{\sqrt{8\pi(1-\bar{\alpha})^3}}\left[0.9e^{-\frac{(x-\sqrt{\bar{\alpha}})^2+(y-\sqrt{\bar{\alpha}})^2}{2(1-\bar{\alpha}(t))}}\left((y-\sqrt{\bar{\alpha}})(y\bar{\alpha}-\sqrt{\bar{\alpha}})-\bar{\alpha}(1-\bar{\alpha})\right)\right.$$

$$+ 0.09e^{-\frac{(x+\sqrt{\bar{\alpha}})^2+(y+\sqrt{\bar{\alpha}})^2}{2(1-\bar{\alpha}(t))}}\left((y+\sqrt{\bar{\alpha}})(y\bar{\alpha}+\sqrt{\bar{\alpha}})-\bar{\alpha}(1-\bar{\alpha})\right)$$

$$\left. + 0.01e^{-\frac{(x-\sqrt{\bar{\alpha}})^2+(y+\sqrt{\bar{\alpha}})^2}{2(1-\bar{\alpha}(t))}}\left((y+\sqrt{\bar{\alpha}})(y\bar{\alpha}+\sqrt{\bar{\alpha}})-\bar{\alpha}(1-\bar{\alpha})\right)\right], \qquad (31)$$

which crucially does not vanish for all $y$. Hence the Fokker-Planck equation that governs the distributions of the forward diffusion process is not satisfied for $p_t^{\text{replace}}(x)$, and replacement sampling will not sample from the true conditional distribution.

## C    REPAINT AS LANGEVIN DYNAMICS

The algorithm that RePaint uses for inpainting is presented in Algorithm 3. We can combine Lines 6 and 9, we can compute the result of inner loop iterations with $u \neq N_{\text{inner}}$ in one update as

$$x_t \leftarrow \sqrt{\frac{1-\beta_{t-\Delta t}}{1-\beta_t}}\left(x_t - \beta_t s_\theta\left(x_t^{(1:n-1)} \oplus \sqrt{\bar{\alpha}(t)}\tilde{x} + \sqrt{1-\bar{\alpha}(t)}z', t\right)\right)$$

$$+ \sqrt{\beta_t(1-\beta_{t-\Delta t})}z + \sqrt{\beta_{t-\Delta t}}z'' \qquad (32)$$

$$\leftarrow \sqrt{\frac{1-\beta(t-\Delta t)\Delta t}{1-\beta(t)\Delta t}}\left(x_t - \beta(t)\Delta t s_\theta\left(x_t^{(1:n-1)} \oplus \sqrt{\bar{\alpha}(t)}\tilde{x} + \sqrt{1-\bar{\alpha}(t)}z', t\right)\right)$$

$$+ \sqrt{\beta(t)\Delta t(1-\beta(t-\Delta t)\Delta t) + \beta(t-\Delta t)\Delta t}z, \qquad (33)$$

where in the final line we have used $\beta_t = \beta(t)\Delta t$ and the identity $a\mathcal{N}(0,I) + b\mathcal{N}(0,I) \sim \sqrt{a^2+b^2}\mathcal{N}(0,I)$. To leading order in $\Delta t$, we now have

$$x_t \leftarrow x_t - \beta(t)\Delta t s_\theta\left(x_t^{(1:n-1)} \oplus \sqrt{\bar{\alpha}(t)}\tilde{x} + \sqrt{1-\bar{\alpha}(t)}z', t\right) + \sqrt{2\beta(t)\Delta t}z, \qquad (34)$$

which is Langevin dynamics with step size $\beta(t)\Delta t$ and target distribution defined implicitly by

$$\nabla \log p_t^{\text{repaint}}(x) = \mathbb{E}_{z \sim \mathcal{N}(0,I)}\left[s_\theta\left(x^{(1:n-1)} \oplus \sqrt{\bar{\alpha}(t)}\tilde{x} + \sqrt{1-\bar{\alpha}(t)}z, t\right)\right]. \qquad (35)$$

Recalling that $s_\theta(x,t)$ is an approximation to the score, we have

$$\lim_{t \to 0}\nabla \log p_t^{\text{repaint}}(x) \approx \nabla \log p_0\left(x^{(1:n-1)} \oplus \tilde{x}\right), \qquad (36)$$

so $p_t^{\text{repaint}}(x)$ anneals to $p_0(x|x^{(n)} = \tilde{x})$.

## D    SCORE FUNCTIONS FOR TRAINING-FREE GUIDANCE

For reference, we list the families of distributions expounded in Sections 3.3 and 3.4 and their corresponding score functions used in Section 4.

---

**Algorithm 3** RePaint for inpainting

---

**Require:** target value $\tilde{x}$, step size $\eta$, number of inner loop iterations $N_{\text{inner}}$
1: $x_T \sim \mathcal{N}(0, I)$
2: **for** $t = T, \ldots, \Delta t$ **do**
3:    **for** $u = 1, \ldots, N_{\text{inner}}$ **do**
4:      $z \sim \mathcal{N}(0, I)$
5:      $z' \sim \mathcal{N}(0, I)$
6:      $x_{t-\Delta t} \leftarrow \frac{1}{\sqrt{1-\beta_t}} \left( x_t - \beta_t s_\theta \left( x_t^{(1:n-1)} \oplus \sqrt{\bar{\alpha}(t)}\tilde{x} + \sqrt{1 - \bar{\alpha}(t)}z', t \right) \right) + \sqrt{\beta_t}z$
7:      **if** $u < N$ **then**
8:        $z'' \sim \mathcal{N}(0, I)$
9:        $x_t \leftarrow \sqrt{1 - \beta_{t-\Delta t}}x_{t-\Delta t} + \sqrt{\beta_{t-\Delta t}}z''$
10:     **end if**
11:    **end for**
12: **end for**
13: **return** $x_0$

---

## D.1 DISTRIBUTIONS

$$p_t^{\text{product}}(x) := \frac{1}{Z_t} p_t(x) \times \mathcal{N}\left(x^{(n)}; \sqrt{\bar{\alpha}(t)}\tilde{x}, 1 - \bar{\alpha}(t)\right) \tag{37}$$

$$p_t^{\text{float}}(x) := \frac{1}{Z_t} p_t(x) \times \sum_{i=1}^{n} \mathcal{N}\left(x^{(i)}; \sqrt{\bar{\alpha}(t)}\tilde{x}, 1 - \bar{\alpha}(t)\right) \tag{38}$$

$$p_t^{\text{perturb}}(x) := \frac{1}{Z_t} p_t(x) \times \sum_{i=1}^{n} \prod_{j \neq i} \mathcal{N}\left(x^{(j)}; \sqrt{\bar{\alpha}(t)}\tilde{x}^{(j)}, 1 - \bar{\alpha}(t)\right) \tag{39}$$

$$p_t^{\text{linear}}(x) := \frac{1}{Z_t} p_t(x) \times \mu \prod_{i=1}^{n} \sum_{j} w_j \mathcal{N}\left(x^{(i)}; \sqrt{\bar{\alpha}(t)}\tilde{x}_j, 1 - \bar{\alpha}(t)\right) \tag{40}$$

$$p_t^{\text{mean}}(x) := \frac{1}{Z_t} p_t(x) \times \frac{\mu \prod_{i=1}^{n} \sum_{j} w_j \mathcal{N}\left(x^{(i)}; \sqrt{\bar{\alpha}(t)}\tilde{x}_j, 1 - \bar{\alpha}(t)\right)}{\prod_{i=1}^{n} \sum_{j \neq \text{PAD}} \mathcal{N}\left(x^{(i)}; \sqrt{\bar{\alpha}(t)}\tilde{x}_j, 1 - \bar{\alpha}(t)\right)} \tag{41}$$

## D.2 SCORE FUNCTIONS

$$\nabla \log p_t^{\text{product}}(x) = \nabla \log p_t(x) - \frac{x^{(n)} - \sqrt{\bar{\alpha}}\tilde{x}}{1 - \bar{\alpha}} \tag{42}$$

$$\nabla \log p_t^{\text{float}}(x) = \nabla \log p_t(x) - \bigoplus_{i=1}^{n} \text{softmax}_i \left( -\frac{(x^{(i)} - \sqrt{\bar{\alpha}}\tilde{x})^2}{2(1 - \bar{\alpha})} \right) \frac{x^{(i)} - \sqrt{\bar{\alpha}}\tilde{x}}{1 - \bar{\alpha}} \tag{43}$$

$$\nabla \log p_t^{\text{perturb}}(x) = \nabla \log p_t(x)$$
$$- \bigoplus_{i=1}^{n} \left( 1 - \text{softmax}_i \left( -\frac{\sum_{j \neq i}(x^{(j)} - \sqrt{\bar{\alpha}}\tilde{x}^{(j)})^2}{2(1 - \bar{\alpha})} \right) \right) \frac{x^{(i)} - \sqrt{\bar{\alpha}}\tilde{x}^{(i)}}{1 - \bar{\alpha}} \tag{44}$$

$$\nabla \log p_t^{\text{linear}}(x) = \nabla \log p_t(x)$$
$$- \mu \bigoplus_{i=1}^{n} \sum_{j} \text{softmax}_j \left( \log w_j - \frac{(x^{(i)} - \sqrt{\bar{\alpha}(t)}\tilde{x}_j)^2}{2(1 - \bar{\alpha}(t))} \right) \frac{x^{(i)} - \sqrt{\bar{\alpha}(t)}\tilde{x}_j}{1 - \bar{\alpha}(t)} \tag{45}$$

$$\nabla \log p_t^{\text{mean}}(x) = \nabla \log p_t(x)$$
$$- \mu \bigoplus_{i=1}^{n} \sum_{j} \text{softmax}_j \left( \log w_j - \frac{(x^{(i)} - \sqrt{\bar{\alpha}(t)}\tilde{x}_j)^2}{2(1 - \bar{\alpha}(t))} \right) \frac{x^{(i)} - \sqrt{\bar{\alpha}(t)}\tilde{x}_j}{1 - \bar{\alpha}(t)}$$

$$-\mu\bigoplus_{i=1}^{n}\sum_{j\neq\text{PAD}}\text{softmax}_{j\neq\text{PAD}}\left(-\frac{(x^{(i)}-\sqrt{\bar{\alpha}(t)}\tilde{x}_j)^2}{2(1-\bar{\alpha}(t))}\right)\frac{x^{(i)}-\sqrt{\bar{\alpha}(t)}\tilde{x}_j}{1-\bar{\alpha}(t)} \quad (46)$$

## E  TEMPERATURE IN THE REVERSE-TIME PROCESS

In Langevin dynamics, and by extension TFG, we can always sample from a tempered version of the distribution by multiplying through by a constant parameter $\tau$, as

$$\nabla\log(p_t(x)^\tau)=\tau\nabla\log p_t(x). \quad (47)$$

By contrast, we cannot simply multiply the score by $\tau$ and sample from the standard reverse-time diffusion process (Du et al., 2023) by simulating

$$x_{t-\Delta t}=\frac{x_t+\tau\beta_t s_\theta(x_t,t)}{\sqrt{1-\beta_t}}+\sqrt{\beta_t}z_t. \quad (48)$$

To see this, observe that Fokker-Planck equation for the tempered distribution is not satisfied, as

$$\frac{\partial}{\partial t}p_t^\tau-\frac{1}{2}\nabla\cdot(xp_t^\tau)-\frac{1}{2}\nabla^2 p_t^\tau$$
$$=\tau p_t^{\tau-1}\left(\frac{\partial}{\partial t}p_t-\frac{1}{2}\nabla\cdot(xp_t)-\frac{1}{2}\nabla^2 p_t\right)+(\tau-1)p_t^\tau-\frac{1}{2}\tau(\tau-1)p_t^{\tau-2}(\nabla p_t)^2$$
$$=(\tau-1)p_t^\tau-\frac{1}{2}\tau(\tau-1)p_t^{\tau-2}(\nabla p_t)^2, \quad (49)$$

which does not vanish in general.

## F  IMAGE INPAINTING TASK DETAILS

For our image inpainting experiments we use `google/ddpm-cifar10-32` from `diffusers` (von Platen et al., 2022) with 100 timesteps. We randomly select 1000 images from the held-out test set, mask the appropriate region, and sample from the model.

We find a perceptual improvement to generated images across all sampling methods when multiplying the output of the predicted score function by a constant factor $\tau$, so we modify each of the baselines to include such a temperature parameter. We fix $\tau=1.1$ for RePaint and MCG, and $\tau=1.2$ for replacement sampling and TFG following hyperparameter tuning in the range $[0.5,2.0]$. For MCG, we use the mean squared error as the conditioning loss and find optimal results with $\gamma=0.011$ following hyperparameter tuning in the range $[0.001,0.1]$. For TFG, we fix the step size $\eta=0.04$ when sampling from both $p_t^{\text{replace}}(x)$ and $p_t^{\text{product}}(x)$ after tuning in the range $[0.001,0.1]$.

## G  PROTEIN MOTIF SCAFFOLDING TASK DETAILS

During training, RFDiffusion is exposed to the motif scaffolding task in a classifier-free setting (80% conditional, 20% unconditional); for sampling, RFDiffusion uses a variant of replacement sampling that does not noise the motif according to diffusion time $t$ but fixes the 3D coordinates throughout the reverse process. To emphasise the difference between sampling methodologies we choose to use the RoseTTAFold denoising network unconditionally across all sampling methods to avoid confounding effects of the task-specific conditioning signal.

We adopt the backbone structure representation of RoseTTAFold (Baek et al., 2021), i.e. $x=(r,z)$, with $z\in\mathbb{R}^3$ the translation and $r\in SO(3)$ the rigid rotation of each of the residues, and we decompose the score vectors into their translation and rotation components, i.e. $s_\theta(x_t,t):=\big(s_\theta(z_t,t),s_\theta(r_t,t)\big)$. We perform the TFG updates following the two-part reverse process introduced in RFdiffusion for the translations and rotations separately (see Algorithm 2 in the supp material of Watson et al. (2023)). Specifically, the translation $z$ updates are performed as outlined in Algorithms 1 and 2. For the rotation component we define our TFG score vectors for $p_t^{\text{product}}(x)$ to be

$$\tau\Big(s_\theta(r_t,t)+\bigoplus_{i=1}^{n}\nabla_{r^{(t)}}\log\mathcal{IG}_{SO(3)}(r_t^{(a_i)},\tilde{r}^{(a_i)},\sigma_t^2)\Big), \quad (50)$$

where we have used the rotation score approximation on SO(3) defined in Watson et al. (2023), $\tilde{r}^{(a_i)}$ the $a_i$-th indexed residue of the protein motifs of total length $n$, allowing for non-contiguous multi-motifs. $\sigma_t^2$ is the unchanged variance schedule from RFdiffusion, and $\tau$ the overall temperature. We introduce a separate translation- and rotation-specific Langevin step size parameters $\eta_z, \eta_r$. The update for the rotation component is then

$$
r_{t-\Delta t} \leftarrow r_t \exp\Big\{\eta_r^2(\sigma_t^2 - \sigma_{t-1}^2)r_t\tau\Big(s_\theta(r_t,t) + \bigoplus_{i=1}^{n} \nabla_{r^{(t)}} \log \mathcal{IG}_{SO(3)}(r_t^{(a_i)}, \tilde{r}^{(a_i)}, \sigma_t^2)\Big)
$$
$$
+ \eta_r \sqrt{\sigma_t^2 - \sigma_{t-1}^2} \sum_{d=1}^{3} \epsilon_d f_d\Big\}, \tag{51}
$$

where $\exp$ is the exponential map on $\mathfrak{so}(3)$, the Lie algebra of $SO(3)$, and $\epsilon_d \sim N(0, I)$ and $f_d$ the chosen orthonormal basis of $\mathfrak{so}(3)$.

The RFDiffusion motif scaffolding benchmark contains 25 motif scaffolding tasks defined by contig strings. We generate 50 backbone designs for each task and use the insilico validation pipeline published by Lin et al. (2024) to attribute design *success*. This pipeline uses the inverse folding method ProteinMPNN (Dauparas et al., 2022) to decode eight sequences from each generated backbone, and ESMFold (Lin et al., 2023) to refold these sequences. For each backbone design, the best refolding structure is considered that with the lowest scRMSD to the original designed backbone. Backbone designs are deemed a *success* if they satisfy pLDDT $> 70$, pAE $< 5$, scRMSD $< 2.0$ Å and motif_bb_rmsd $< 1.0$ Å. To assess the diversity of designed backbones we also report the number of *unique successes*, which is determined using single linkage hierarchical clustering to group structures based on structural similarity (TM-score).

We note that the reimplementation of the original RFDiffusion benchmark includes minor differences, particularly the motifs specified by tasks 6exz, 6e6r and 5trv are shifted by one, one and two residues respectively. We choose to match our generations to the Lin et al. (2023) implementation since this allows us to reuse their published evaluation pipeline.

For TFG, we fix $\tau = 1.75$, $\eta_z = 0.05$, and $\eta_r = 0.005$ across the experiments on protein structures following hyperparameter tuning of values in the ranges $[1.0, 2.0]$, $[0.001, 0.1]$, and $[0.001, 0.1]$ respectively. For replacement sampling and RePaint we fix $\tau = 1.5$ following tuning in the range $[1.0, 2.0]$.

## H  FLOATING IMAGE INPAINTING TASK DETAILS

For our floating inpainting experiment, we use `google/ddpm-celebahq-256` from `diffusers` (von Platen et al., 2022) with the number of timesteps fixed to 100.

We peform sampling with TFG on $p_t^{\text{float}}(x)$ on 1000 images across the four quadrants – that is, we generate 4000 total samples. We declare that TFG has selected the correct quadrant if the mean squared error between the original condition and each of the four quadrants of the generated sample is minimised by the same from which the condition is originally taken. Following hyperparameter tuning in the ranges $[0.001, 0.1]$ and $[1.0, 2.0]$, we fix $\eta = 0.025$ and $\tau = 1.2$.

## I  MULTI-MOTIF SCAFFOLDING TASK DETAILS AND FURTHER RESULTS

We use the same experimental setup and hyperparameters described in Section 4.1, and sample from RFDiffusion to scaffold the multi-motif. For a total number $M$ possible locations of a motif of length $n$, the TFG rotation score vector for $p_t^{\text{float}}(x)$ is defined to be

$$
\tau\Big(s_\theta(r_t,t) + \sum_{m=1}^{M} \lambda_m \bigoplus_{i=1}^{n} \nabla_{r^{(t)}} \log \mathcal{IG}_{SO(3)}(r_t^{(a_i)}, \tilde{r}_m^{(a_i)}, \sigma_t^2)\Big), \tag{52}
$$

where $a_i^m$ indexes the $i$-th residue index of the $m$-th motif $(\tilde{z}_m, \tilde{r}_m)$, and

$$
\lambda_m = \text{softmax}_m\Big(-\frac{(z_t - \sqrt{\bar{\alpha}}\tilde{z}_m)^2}{2(1-\bar{\alpha})}\Big), \tag{53}
$$

where inner product is defined over each motif $m$ residue indices and across the 3D coordinates. Note that for simplicity, we have defined the coefficients $\lambda_m$ only in terms of the translation components, ignoring the rotation distribution entirely. In practice we have found that this simplification is sufficient for guiding the motif to preferable locations.

In our experiment, we select the PDB ID `1bcf` from the RCSB Protein Data Bank (Armstrong et al., 2020) and define a multi-motif by the residue maps A112-125 and A129-134. The separation of the two motifs is chosen to be 3 amino-acid residues long. We define the set of $M = 5$ candidate motif locations with separations of $(3, 9, 15, 21, 28)$ amino-acids residues and the task is to recover the original separation (i.e. 3) by sampling from $p_t^{\text{float}}(x)$ with score defined in Equation 52. We perform 100 motif scaffolding tasks for $p_t^{\text{float}}(x)$, $p_t^{\text{product}}(x)$, and replacement sampling, where for the latter two methods, we perform sampling with maps A112-125/10-20/A129-134/10-20, demonstrating the consequence of mis-specification of the inter-motif gap sizes.

As shown in the main text, the freedom to place the motif leads higher $\text{pLDDT}_{\text{RF}}$ scores. Here we use the pLDDT score directly from the RoseTTAFold denoising model. We also examine the coefficients $\lambda_m$. As shown in Figure 8, the score modification as given in Equation 53 drives the motif to its eventual position after $\sim 25$ (out of $T = 50$) reverse timesteps and the TFG with $p_t^{\text{float}}(x)$ strongly favors the original inter-motif gap size of 3, which coincides with higher confidence structures.

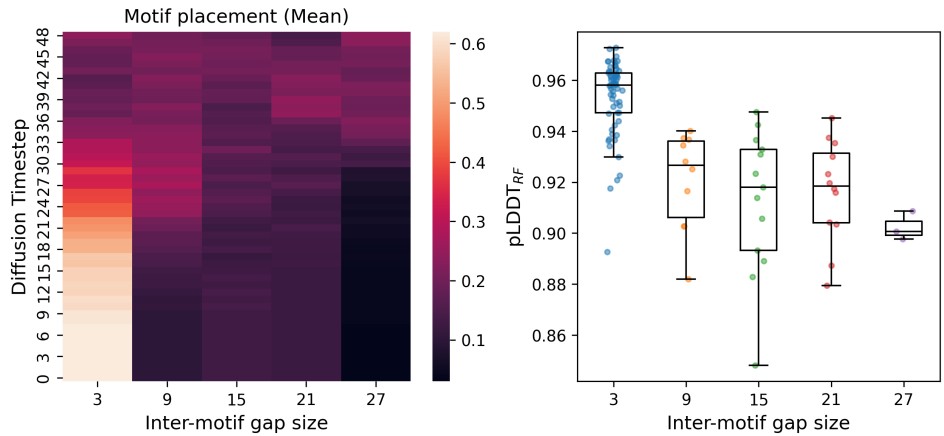

Figure 8: (Left) Mean motif placement probabilities $\lambda_m$ (Equation 53) over 100 tasks across the diffusion process. (Right) The $\text{pLDDT}_{\text{RF}}$ scores for structures as a function of inter-motif gap size.

## J  CDR3$\beta$ SEQUENCE DIFFUSION MODEL

We collect CDR3$\beta$ sequences from three public databases: VDJdb (Shugay et al., 2018), IEDB (Vita et al., 2019), and McPAS-TCR (Tickotsky et al., 2017). After removing duplicates, 89928 unique sequences remain. Sequence are converted to points in Euclidean space by one-hot encoding the amino acid sequence. We corrupt the encodings by adding Gaussian noise according to a continuous-time noise schedule, where $\bar{\alpha}(t)$ is sampled from $\mathcal{N}(-1.3, 1)$, which was chosen as to increase the sampling importance where the .

The denoising model is a modification a transformer-like architecture (Vaswani et al., 2017) for sequence-level proteins, ESM-2 (Lin et al., 2023). We modify the architecture of ESM-2 so that the noise level $\bar{\alpha}(t)$ is appended to the amino acid encoding, and the model head is replace with a linear layer that predicts the noise. We train from scratch with a learning rate of $3 \times 10^{-4}$. We perform two learning rate rescalings by 0.1 after the score matching loss has plateaued for five epochs.

## K  CDR3$\beta$ SEQUENCE TASK DETAILS

For all experiments we use the same unconditional model with training procedure described in Appendix J. For all sampling methods, we select timepoints distributed according to a probability

density function proportional to the normal distribution $\mathcal{N}(-1.3, 1)$, i.e. the distribution used for importance sampling of timepoints during training. For TFG we fix $\eta = 0.001$ (following hyperparameter tuning in the range $[0.001, 0.1]$) and $N_{\text{inner}} = 5$.

