# OpenReview forum: "Training-free guidance of diffusion models for generalised inpainting"
_ICLR.cc/2025/Conference — ICLR 2025 Conference Withdrawn Submission_

### Official Review · Reviewer_egZZ · 2024-10-18

**Soundness:** 2
**Presentation:** 1
**Contribution:** 2
**Rating:** 3
**Confidence:** 3

**Summary:**

The authors propose 'training-free guidance,' a method to sample from many different sorts of conditional distributions. This approach namely applies to inpainting and 'generalized inpainting.' The authors incorporate annealed Langevin Dynamics into the reverse diffusion process, leveraging a condition-forcing term that encourages the sample to converge to the correct conditional distribution. The authors show their approach on image inpainting and protein structure and sequence generation.

**Strengths:**

S1) The authors show good quantitative performance on all experiments, improving over the SOTA.

S2) The authors provide nice theory both in the paper and in appendices to back up their claims.

**Weaknesses:**

W1) There is no differentiation between scalar and vector quantities. Please fix this; it makes reading and following the math in the paper harder than it needs to be. One solution: bold vector quantities. I think that this is unacceptable.

W2) The characterization of the diffusion literature in Sections 1 and 2 is inaccurate to me, as well as how the paper is positioned in it. Some examples:
- The opening sentence states that DDPMs have gained popularity. This is inaccurate. Diffusion models at large have gained popularity, DDPM is *one specific discretization* of the broader diffusion SDE (for the variance-preserving case). However, plenty of papers consider the variance-exploding (VE) SDE, which is not a 'DDPM.'
- The discussion of the diffusion SDE does not cite Song et al.'s seminal 2021 work "Score-Based Generative Modeling through Stochastic Differential Equations." This paper is cited later, but in an off-hand way when discussing annealed Langevin Dynamics. This needs to be the centerpiece around which you build Section 2. You also should discuss the variance-preserving (VP) SDE specifically (you do this, you just don't correctly characterize it), and note that DDPM is one specific discretization of it.
- I am not satisfied with the other diffusion works discussed. Based on Sections 1 and 2, you would think that there are barely any diffusion methods out there doing conditional generation! A more robust discussion of related work should be included, even if it is in an appendix.

W3) I think that you fail to accurately characterize the method. You talk about DDPM, but the method shown in Algs. 1 and 2 is not a DDPM sampler. It is solving the VE-SDE and is effectively a conditional version of [1]. That's not a problem, but it is problematic that things are not described well.

W4) The paper assumes too much knowledge about the very niche protein-based experiments. As someone with a background in imaging problems, I found the protein experiments hard to follow. There is not a clear description of the problems, and vocabulary that a layman is not familiar with is used to describe things. For instance: what is a structural motif? I could google it, but I should not have to while reading your paper. Since ICLR is a general conference, there will likely be many potential readers who know nothing about this field. Additional details in the paper, or even an appendix, would benefit this paper greatly.

W5) This is minor, but I find the use of 'training-free' to be a bit misleading. Most methods for solving conditional problems with pre-trained unconditional diffusion models are training-free. This is part of why diffusion models have become so popular: their versatility. I don't think there is anything actionable here, and this point does not impact my score, but I wanted to point it out.

[1] Song & Ermon, "Generative Modeling by Estimating Gradients of the
Data Distribution," 2020

**Questions:**

See weaknesses.

**Details Of Ethics Concerns:**

There are no ethical concerns.

---

### Official Review · Reviewer_gBXR · 2024-10-31

**Soundness:** 2
**Presentation:** 2
**Contribution:** 1
**Rating:** 3
**Confidence:** 5

**Summary:**

The paper proposes several annealing families for solving "generalised inpainting" problems, relying on two main families, namely one based on the replacement method and other based on the product method. They propose sampling from such annealed distributions using sequential Unadjusted Langevin algorithms. They show that the proposed annealed solutions work reasonably on different inpainting tasks.

**Strengths:**

The paper is clearly written and well explained. In particular, the toy problem example is illuminating as the explanation of repaint. The floating inpainting application is interesting and new.

**Weaknesses:**

My main concern with the paper is that it overlooks big chunks of the literature. This makes the paper overstate its novelty aspect.

Sampling from a sequence of distributions using the learned score is known for quite some time. Indeed, the very foundational paper of the current score based generative model, [1], proposes already a Langevin approach from the learned scores. The whole field of score based generative models and all the developments that happened after [1] such as [2], [3] and [4] are intended to precisely remove the need to resort to Langevin and pass directly from $p_{t}$ to $p_{t-1}$.

This is precisely the reason why all the current posterior sampling approaches such as DPS [5] but also all the recent developments are attempts to obtain a similar backward path distribution that do not resort to Langevin. Therefore, reintroducing Langevin seems at the same time straightforward and not relevant.

Furthermore, both $p_{t}^{replace}$ and  $p_t^{product}$ distributions defined in eq(8) and eq(14) are known in the literature as well as the fact that they do not correspond to an equivalent backward of a forward diffusion:

 * $p_{t}^{replace}$ corresponds to the intermediate distributions from [10].
 * $p_t^{product}$  corresponds to the intermediate distributions proposed in [6] for the inpainting case (i.e. eq. 2.3 from [6]).

In both [10] and [6] the authors propose using Sequential Monte Carlo (SMC). Of course, the extension to using sequential Unadjusted Langevin is straightforward. Both sequential ULA are SMC are asymptotically correct samplers. Therefore, the paper should not overlook [10] and [6] and present their approach as changing SMC to ULA in [10] and [6] and why this is relevant (either theoretically or numerically).

In general, the paper lacks comparison to the standard literature. Much of the current state of the art approaches not being considered, such as [7], [8] and [9].

For table 1, while the shown metrics (MSE) and LPIPS are standard in the literature, their usage in extremely ill-conditioned inverse problems (as the one described in the image inpainting section) are not relevant. Indeed, for such problems I would suggest showing the proposed method performs well in a case where sampling from the reference distribution is feasible and use those to calculate some distribution related distance, such as the wasserstein distance. Indeed, we can see for example that in Figure 2, the second column for the Top inpaiting that it is hard to discard that any of the proposed images are from the actual posterior of the associated inpainting problem. This is an illustration that comparing those to the "real image" is not at all a good proxy to the performance of such methods.

Minor comments:
The term temperature is not defined in the main text. While people familiar with Langevin algorithms can understand directly, it should be defined in the main paper as it is used in figure 3.


[1] Song, Y., & Ermon, S. (2019). Generative modeling by estimating gradients of the data distribution. Advances in neural information processing systems, 32.
[2] Ho, J., Jain, A., & Abbeel, P. (2020). Denoising diffusion probabilistic models. Advances in neural information processing systems, 33, 6840-6851.
[3] Song, Y., Sohl-Dickstein, J., Kingma, D. P., Kumar, A., Ermon, S., & Poole, B. Score-Based Generative Modeling through Stochastic Differential Equations. In International Conference on Learning Representations.
[4] Song, J., Meng, C., & Ermon, S. Denoising Diffusion Implicit Models. In International Conference on Learning Representations.
[5] Chung, H., Kim, J., Mccann, M. T., Klasky, M. L., & Ye, J. C. Diffusion Posterior Sampling for General Noisy Inverse Problems. In The Eleventh International Conference on Learning Representations.
[6] Cardoso, G., Y.Janati, Le Corff, S., & Moulines, E. (2023). Monte Carlo guided Denoising Diffusion models for Bayesian linear inverse problems. In The Twelfth International Conference on Learning Representations.
[7] Wu, L., Trippe, B., Naesseth, C., Blei, D., & Cunningham, J. P. (2024). Practical and asymptotically exact conditional sampling in diffusion models. Advances in Neural Information Processing Systems, 36.
[8]Janati, Y., Durmus, A., Moulines, E., & Olsson, J. (2024). Divide-and-Conquer Posterior Sampling for Denoising Diffusion Priors. arXiv preprint arXiv:2403.11407.
[9] Mardani, M., Song, J., Kautz, J., & Vahdat, A. A Variational Perspective on Solving Inverse Problems with Diffusion Models. In The Twelfth International Conference on Learning Representations.
[10] Trippe, B. L., Yim, J., Tischer, D., Baker, D., Broderick, T., Barzilay, R., & Jaakkola, T. S. Diffusion Probabilistic Modeling of Protein Backbones in 3D for the motif-scaffolding problem. In The Eleventh International Conference on Learning Representations.

**Questions:**

What is the run time and the NFE (neural function evaluation number) for each of the algorithms in table 1?

---

### Official Review · Reviewer_FZxP · 2024-11-01

**Soundness:** 2
**Presentation:** 1
**Contribution:** 2
**Rating:** 3
**Confidence:** 3

**Summary:**

In this paper, the authors introduce training-free guidance, an inpainting method using pre-trained conditional generative models. The framework has been applied on images (CIFAR10 dataset) and protein structures (RFdiffusion dataset). In the toy example, the authors show how sampling from a pre-trained conditional model would give a bias based on the inbalanced training set, leading to worse inpainting output using directly the inference denoising process directly to inpaint the missing area. The authors propose a new sampling method based on annealed Langevin Dynamics to sample exact sampling for generalised inpainting conditions.

**Strengths:**

I like the idea to show the performance on a method simultaneously to different data distributions like images and proteins. I'm more an expert on the imaging part so I would leave more space to the other reviewers regarding the protein part.

**Weaknesses:**

Following the story of the paper seems pretty hard. I've understood the problem that they want to solve but all the sections are independent and there isn't a continuous flow in the narrative of the paper. It took me several time to understand properly what is the solution that the authors provided to the task that they are trying to solve (task that is already well-known in literature). The sections names are understandable but when they describe the approach they fall into details without providing the full picture on the work. I suggest to the authors to revisit the overall writing of the paper. Following the paper I'm

-this is the DDPM background (2.1)

-this is the annealed Langevin dynamics (2.2)

-the inpainting problem is sampling part of x, fixing a subset M of it
these (3.1)

-inpainting would lead to bias due to unbalanced data in the training set (3.2)

I feel that these 4 sections are independent between each other and they don't give an answer to the claims in the following sections:

-line 165: "Notice that although the family of distributions p_t^replace(x(1:n−1)) does not correspond to a forward diffusion process, it does anneal to the desired distribution p0(x|x(n) = x ̃), so we are at liberty to use annealed MCMC approaches.", where does this claim come from?

-line 172: "The key insight that enables us to generalise TFG to a much wider range of tasks is that we do not have to use preplace(x(1:n−1)) as t our annealed family. Instead, we can define any family of distributions – including a family that can be adapted for generalised inpainting problems. To this end, we define a new family of distributions ...", also this one what's the actual claim to get to this result?

From here they provide use-cases based on this claims. There are experiments and ablations studies.

Talking about the technical part of the work, I feel like that some definitions and key elements are missing. For example how to get to equations (10) is not clear for me. Like what's the relation of the probability on the fixed dimensions M (conditional part) and the other. Indeed, in equation 10, from equality it seems that the two are related in same way. This vanishing part is not well explained as other key parts of the paper. Working also with the DDPM which is a generalisation of the score-based model, claiming that the model has some key issues in the inpainting part from a mathematical perspective sounds a bit like finding an issue on the edge. I would like to see the mathematical description directly on the broader use case.

**Questions:**

The paper is not really clear in a lot of points. I'd ask to the authors to add more descriptions on several part of the work and guide more the reader to the true impact of the work.

---

### Official Review · Reviewer_2JGn · 2024-11-02

**Soundness:** 3
**Presentation:** 3
**Contribution:** 4
**Rating:** 8
**Confidence:** 3

**Summary:**

This paper enhances inpainting and outpainting in generative diffusion models without requiring additional training, making it likely to be impactful within the diffusion community as an out-of-the-box improvement. This is achieved with annealed Langevin dynamics, ensuring convergence to the exact conditional distribution, unlike previous heuristic-based methods.

**Strengths:**

* The main strength is that it's *training-free* while demonstrating quantitatively and qualitatively improved inpainting and outpainting with pretrained unconditional models.
   * This is a significant contribution to the community, given the importance of inpainting applications in real-world settings.
* The paper demonstrates both image and protein applications, with strong results in both cases.
* The approach is original and theoretically justified, while being easy to understand, with a simple toy example.

**Weaknesses:**

* Several real-world inpainting image solutions, such as Fooocus in the StableDiffusion community, replace masked inpainting pipelines with soft masking approaches, e.g., Differential Diffusion (Levin et al., 2023) and DiffEdit (Couairon et al., ICLR 2023). While I appreciate these are not quite the same, I think there should be relevant discussion or a simple experiment/demonstration of this setting, if applicable.
* I think the paper could be strengthened with more examples such as Table 2, but on higher-resolution datasets. In particular, it's difficult to see where this improves over MCG on CIFAR10 at $32^2$. Identifying a larger pretrained unconditional model, such as on FFHQ $1024^2$ or equivalent, and evaluating this would be more convincing if the approach works well in that setting.
* The discussions on 3.5 RePaint seemed slightly tangential, and the narrative felt disconnected during my first reading of these extensions, making it difficult to follow. I think there should be a brief summary at the end of 3.4 that wraps up these parts of the methodology.

**Questions:**

* Would this framework be applicable to Flow Matching (Lipman et al., ICLR 2023), potentially offering improved results with straight paths through the transition space, or would TFG require significant modification in this setting?
* Is it possible to show NLLs derived from the conditionally generated samples, for example compared against NLL values for fully unmasked samples from the test set? I'm assuming this is possible following Song et al., given that this is fundamentally a pretrained unconditional model.
* Could you clarify the gray bleeding/discontinuity of the mask in Figure 2, top-right image for TFG (product)? It seems to occur for this ship example but not for the horse example.

---

### Official Review · Reviewer_c221 · 2024-11-03

**Soundness:** 2
**Presentation:** 2
**Contribution:** 2
**Rating:** 3
**Confidence:** 4

**Summary:**

The authors propose a method to turn unconditional diffusion models into conditional models by using a combination of replacement sampling and Langevin dynamics. Replacement sampling, as defined by the authors, does not sample from the correct conditional distribution $p_\theta(x^{unobserved} \mid x^{observed})$, therefore the authors define a target distribution, with the prior provided by the unconditional model and a likelihood for the observations:

$p_\theta(x_t) N(x^{observed}_t; \sqrt{\bar{\alpha}(t)}x^{observed}, 1 - \bar{\alpha}(t))$

and run an inner loop after every de-noising step done with replacement sampling.

The authors then generalize in-painting and define different likelihoods and show that their algorithm involving replacement sampling and Langevin sampling can be applied to these problems.

**Strengths:**

The authors provide a motivation for their problems with a toy problem as well as theoretical justifications of their methods and shortcomings of existing methods. The authors also conduct a thorough set of experiments and include some of the appropriate baselines.

**Weaknesses:**

Section 3.1

The definition of $p^{replace}_t(x^{\not\in M})$ is ambiguous:

1. What does the super-script here $p_t()^{\not\in M}$ mean?
2. Are the authors defining a conditional distribution here?  Otherwise $p^{replace}_t$ is not a density that integrates to 1.

Regarding the Fokker-Planck based analysis of the $p^{replace}_t$

1. Equation 10 on lines 116-122 are stated without any derivation
2. The term $\partial_t p^{replace}_t - 0.5 \nabla_{x \not\in M} (x^{\not\in M} p_t^{replace}) - \frac{1}{2}\nabla^2_{x \not\in M}p_t^{replace}$ is made equal to another set of terms without any proof, in either the main text or the appendix.
3. Moreover, Laplacians are not additive, $\Delta_{x^M} p_t + \Delta_{x^{\not\in} M} p_t \neq \Delta_x p_t$, since the second order gradients that interact between $x^M$ and $x^{\not\in M}$ do not show up in the left hand side.
4. Applying eq 10 to the analysis of the toy problem makes that analysis unclear as well.

In the toy example described in section 3.2:

1. The data distribution is defined as a mixture of atoms, i.e. a discrete distribution, however in the first three panels of Figure 1, the distribution seems to be a mixture of Gaussians.
2. The authors do not explain how the marginal distribution biases the replacement sampling algorithm.

Comments on the algorithms 1 and 2

1. For the product algorithm, can the authors provide any analysis of sampling from that particular product distribution ? Does that fix the bias present in the replacement sampling method?
2. In both algorithm 1, the authors instead of sampling from a high-dimensional isotropic Gaussian, use the mean of the Gaussian as the sample. While in low-dimensions, this can be fine, in high-dimensions the samples of a Gaussian lie on a shell centered around the mean. Making the mean, or an epsilon ball around it, an increasingly atypical sample as the dimensions increase [Vershynin 2018]. For instance, $E[|| x ||_2^2] = d E[x^2_i] = d (1 - \bar{\alpha}(t))$ where $x \sim N(0, 1 - \bar{\alpha}(t) I_d)$.


The authors claim that the product distributions in section 3.3 as a novel contribution, however:

1. [Wu et al 2024] also define the same likelihood in eq 13, implying sampling from a similar posterior distribution as proposed by the authors. [Wu et al 2024] use SMC for sampling instead of Langevin sampling. However, in [Wu et al 2024] the authors provide a proof for exact sampling.

**Related work.** There are other approaches which provide fixes for the bias in replacement sampling. For instance,

1. Practical and Asymptotically Exact Conditional Sampling in Diffusion Models [Wu et al 2024]
2. RePaint+  [Rout et al 2023]

however, the authors do not engage with these works.

**Minor points.**

1. the authors use the term $\oplus$, without defining it in eq 7.
2. What does the super-script, $p_t()^{\not\in M}$ in eq 8 mean?
3. the use of $\nabla^2$ to denote the Laplacian in eq 9 is incorrect, it should be either $\Delta$ or $\nabla \cdot (\nabla p)$.
4. In section 3.5, the repaint interlude, the authors introduce the distribution $p^{repaint}_t$ without defining it

References

[Wu et al 2024] Wu, Luhuan, Brian Trippe, Christian Naesseth, David Blei, and John P. Cunningham. "Practical and asymptotically exact conditional sampling in diffusion models.”  *Advances in Neural Information Processing Systems* 36 (2024).

[Rout et al 2023] Rout, Litu, Advait Parulekar, Constantine Caramanis, and Sanjay Shakkottai. "A theoretical justification for image inpainting using denoising diffusion probabilistic models." *arXiv preprint arXiv:2302.01217* (2023).

[Veryshnin 2018] Vershynin, R., 2018. *High-dimensional probability: An introduction with applications in data science* (Vol. 47). Cambridge university press.

**Questions:**

See the weakness section.

The main questions are:
1. The derivations for the Fokker-Planck analysis look incomplete
2. The notations complicate understanding the derivations and the main text.
3. An analysis of the proposed product distribution is also missing. It is not clear whether Langevin sampling from that target distribution will sample from the correct conditional distribution. See [wu et al 2024] for an analysis of sampling a proposed posterior distribution which yields the correct conditional distribution.

3. The version of replacement sampling where the authors use the mean instead of a random sample from an isotropic Gaussian is not the standard implementation of replacement sampling, unless the authors can provide a citation.

---

### Comment · Area_Chair_WZjP · 2024-11-25
**Authors' Rebuttal**

Dear Authors,

As the author-reviewer discussion period is approaching its end, I strongly encourage you to read the reviews and engage with the reviewers to ensure the message of your paper has been appropriately conveyed and any outstanding questions have been resolved.

This is a crucial step, as it ensures that both reviewers and authors are on the same page regarding the paper's strengths and areas for improvement.

Thank you again for your submission.

Best regards,

AC

---

### Note · Authors · 2024-11-26

**Comment:**

We would like to thank all the reviewers and the A/C for their time and effort put into reviewing our manuscript, and for the many helpful suggestions for improvement. Whilst there was a broad appreciation of the applicability of our work across multiple domains, the novelty of some of the experiments and the clarity of our illustrative toy example, we acknowledge the concerns that our submission has not made a thorough comparison to the recent literature, which unfortunately overshadowed many aspects of our sampling proposal around generalisability. In light of the time needed for a proper revision, we have made the decision to withdraw our paper.

**Withdrawal Confirmation:**

I have read and agree with the venue's withdrawal policy on behalf of myself and my co-authors.